# Uptake of infant and preschool immunisations in Scotland and England during the COVID-19 pandemic: An observational study of routinely collected data

Fiona McQuaid[1]*, Rachel Mulholland[2], Yuma Sangpang Rai[3], Utkarsh Agrawal[4], Helen Bedford[5], J. Claire Cameron[6], Cheryl Gibbons[6], Partho Roy[3], Aziz Sheikh[2], Ting Shi[2], Colin R. Simpson[2,7], Judith Tait[6], Elise Tessier[3], Steve Turner[8], Jaime Villacampa Ortega[6], Joanne White[3], Rachael Wood[2,6]

1 Department of Child Life and Health, University of Edinburgh, Edinburgh, United Kingdom, 2 Usher Institute, University of Edinburgh, Edinburgh, United Kingdom, 3 Immunisation and Countermeasures Division, National Infections Service, Public Health England, London, United Kingdom, 4 School of Medicine, University of St Andrews, St Andrews, United Kingdom, 5 UCL Great Ormond Street Institute of Child Health, London, United Kingdom, 6 Public Health Scotland, Glasgow and Edinburgh, United Kingdom, 7 School of Health, Victoria University of Wellington, Wellington, New Zealand, 8 Women and Children Division, NHS Grampian, Aberdeen, United Kingdom

* Fiona.mcquaid@ed.ac.uk

**Data Availability Statement:** All relevant data are within the manuscript and its Supporting

## Abstract

### Background

In 2020, the SARS-CoV-2 (COVID-19) pandemic and lockdown control measures threatened to disrupt routine childhood immunisation programmes with early reports suggesting uptake would fall. In response, public health bodies in Scotland and England collected national data for childhood immunisations on a weekly or monthly basis to allow for rapid analysis of trends. The aim of this study was to use these data to assess the impact of different phases of the pandemic on infant and preschool immunisation uptake rates.

### Methods and findings

We conducted an observational study using routinely collected data for the year prior to the pandemic (2019) and immediately before (22 January to March 2020), during (23 March to 26 July), and after (27 July to 4 October) the first UK "lockdown". Data were obtained for Scotland from the Public Health Scotland "COVID19 wider impacts on the health care system" dashboard and for England from ImmForm.

Five vaccinations delivered at different ages were evaluated; 3 doses of "6-in-1" diphtheria, tetanus, pertussis, polio, *Haemophilus influenzae* type b, and hepatitis B vaccine (DTaP/IPV/Hib/HepB) and 2 doses of measles, mumps, and rubella (MMR) vaccine. This represented 439,754 invitations to be vaccinated in Scotland and 4.1 million for England. Uptake during the 2020 periods was compared to the previous year (2019) using binary logistic regression analysis. For Scotland, uptake within 4 weeks of a child becoming eligible by age was analysed along with geographical region and indices of deprivation. For Scotland and

Information files and further Scottish data can be accessed via the Public health Scotland wider impacts dashboard https://scotland.shinyapps.io/phs-covid-wider-impact/. The English data used is included in the supplementary information. Any further queries regarding the data on ImmForm should be directed to EarlyChildhoodBaseline@phe.gov.uk. All code used in the analyses is publicly available via the EAVE II GitHub page (https://github.com/EAVE-II).

**Funding:** This analysis was part of the EAVE II project. EAVE II is funded by the Medical Research Council (MC_PC_19075), https://mrc.ukri.org/, with the support of BREATHE: the Health Data Research Hub for Respiratory Health (MC_PC_19004), https://www.hdruk.ac.uk/helping-with-health-data/health-data-research-hubs/breathe, which is funded through the UK Research and Innovation Industrial Strategy Challenge Fund and delivered through Health Data Research UK. The funders had no role in study design, data collection and analysis, decision to publish, or preparation of the manuscript.

**Competing interests:** I have read the journal's policy and the authors of this manuscript have the following competing interests: AS is a member of the Scottish Government Chief Medical Officer's COVID-19 Advisory Group. AS is a member of the Editorial Board of PLOS Medicine. HB is a member of the NICE committee developing guidance on increasing vaccine uptake. CRS declares funding from the Medical Research Council, the National Institute for Health Research, Chief Scientist Office, and New Zealand Ministry for Business, Innovation and Employment and Health Research Council during the conduct of this study. All other authors declare no competing interests.

**Abbreviations:** CI, confidence interval; COVID-19, Coronavirus Disease 2019; DTaP/IPV/Hib/HepB, diphtheria, tetanus, pertussis, polio, *Haemophilus influenzae* type b, and hepatitis B; HSCP, Health and Social Care Partnership; MMR, measles, mumps, and rubella; OR, odds ratio; SARS-CoV-2, Severe Acute Respiratory Syndrome Coronavirus 2; SIMD, Scottish Index of Multiple Deprivation; WHO, World Health Organization.

England, we assessed whether immunisations were up-to-date at approximately 6 months (all doses 6-in-1) and 16 to 18 months (first MMR) of age.

We found that uptake within 4 weeks of eligibility in Scotland for all the 5 vaccines was higher during lockdown than in 2019. Differences ranged from 1.3% for first dose 6-in-1 vaccine (95.3 versus 94%, odds ratio [OR] compared to 2019 1.28, 95% confidence intervals [CIs] 1.18 to 1.39) to 14.3% for second MMR dose (66.1 versus 51.8%, OR compared to 2019 1.8, 95% CI 1.74 to 1.87). Significant increases in uptake were seen across all deprivation levels.

In England, fewer children due to receive their immunisations during the lockdown period were up to date at 6 months (6-in-1) or 18 months (first dose MMR). The fall in percentage uptake ranged from 0.5% for first 6-in-1 (95.8 versus 96.3%, OR compared to 2019 0.89, 95% CI 0.86– to 0.91) to 2.1% for third 6-in-1 (86.6 versus 88.7%, OR compared to 2019 0.82, 95% CI 0.81 to 0.83).

The use of routinely collected data used in this study was a limiting factor as detailed information on potential confounding factors were not available and we were unable to eliminate the possibility of seasonal trends in immunisation uptake.

## Conclusions

In this study, we observed that the national lockdown in Scotland was associated with an increase in timely childhood immunisation uptake; however, in England, uptake fell slightly. Reasons for the improved uptake in Scotland may include active measures taken to promote immunisation at local and national levels during this period and should be explored further. Promoting immunisation uptake and addressing potential vaccine hesitancy is particularly important given the ongoing pandemic and COVID-19 vaccination campaigns.

## Author summary

### Why was this study done?

- Early reports from multiple countries suggested that the SARS-CoV-2 (COVID-19) pandemic and associated control measures such as "lockdowns" could be detrimental to routine childhood immunisation uptake.

- A fall in the number of children receiving their routine immunisations could leave the population vulnerable to multiple infectious diseases such as measles.

- However, these reports generally only assessed the immediate effect within a few weeks of national lockdowns, and the longer-term impact of COVID-19 control measures on routine childhood immunisation uptake requires further evaluation.

- It is important to understand how the pandemic affected the uptake of routine childhood immunisations in order to discover what factors influence immunisation uptake and help plan for the future.

## What did the researchers do and find?

- Information on childhood immunisation uptake is routinely collected in Scotland and England. Because of concerns about the effect of the pandemic on immunisation uptake, this information was collected in more detail and we used this to examine what happened over the lockdown period.

- To see if there was any change in immunisation uptake, we compared the information from immediately before, during, and after the 2020 lockdown in the United Kingdom to information from the previous year (2019).

- We found that immunisation uptake in Scotland rose significantly across lockdown, with over 7,000 more children receiving their immunisations on time compared to the previous year. In England, there was a slight fall in uptake.

## What do these findings mean?

- These findings suggest that, despite early concerns, infant and preschool immunisation uptake increased in Scotland over the lockdown period.

- We do not yet have enough information to determine the reasons behind this positive change in Scotland or why uptake fell in England.

- The next step is to assess what factors may have led to this apparent increase in uptake, which could improve uptake of infant and preschool immunisations beyond the pandemic.

- Due to the type of information used, we were unable to fully explore the possibility of immunisation uptake varying by the time of year, and some potentially important details, such as ethnicity, were not available.

## Introduction

The SARS-CoV-2 (COVID-19) pandemic and associated control measures such as national "lockdowns", involving varying restrictions on leaving the home, work, and socialising, have had a profound impact on daily life and the delivery of healthcare worldwide. In the United Kingdom, a national lockdown was announced on 23 March 2020, with instructions that people should only leave their home for a limited number of "essential" reasons [1]. This was accompanied by the reconfiguration of acute healthcare services to support the anticipated influx of COVID-19 patients, cancellation of most elective activity, and pausing of screening programmes [2]. During lockdown, there was evidence of a change in healthcare-seeking behaviour—for example, in Scotland, the uptake of both emergency and elective hospital-based care dropped substantially over the lockdown period [3]. However, within child health, key routine services including childhood immunisations continued through the UK [4].

It has become increasingly apparent that younger children are at low risk of severe disease due to Severe Acute Respiratory Syndrome Coronavirus 2 (SARS-CoV-2) [5] and may be less susceptible to infection with the virus [6]. Yet the wider impact of the pandemic on children in terms of education, mental and physical health, and safeguarding is not yet fully understood

and is likely to be profound [7,8]. One particular area of concern early in the lockdown period was the potential effect on the uptake of routine childhood immunisations [9]. Maintaining high population vaccine coverage is vital for both direct and indirect (via herd immunity) protection against non-COVID-19 infectious diseases.

In July 2020, the World Health Organization (WHO) warned of a potential decline in routine immunisation rates associated with the COVID-19 pandemic, citing a poll from May 2020 in which respondents from 82 countries suggested disruption to immunisation programmes was widespread [10]. Initial reports from England [11], Pakistan [12], South Africa [13], Singapore [14], and the USA [15] were concerning, suggesting a fall in children receiving their scheduled vaccinations in the very early weeks of national lockdowns, though, to the best of our knowledge, the full impact has not yet been assessed. However, the English and US studies relied on surrogate measures of vaccine uptake; number of vaccines delivered/ordered (without a corresponding denominator) [11,15] and the Singaporean study used convenience sampling with multiple assumptions for missing data [14]. Longer-term data were available from the KwaZulu-Natal Province of South Africa [13] and Sindh Province of Pakistan [12], which appeared to show some recovery after an initial fall in uptake; however, in Pakistan, part of the lockdown restrictions involved shutting down of outreach immunisation programme. The overall impact of lockdown on immunisation uptake in higher-income countries, which maintained their routine immunisation programmes, is unclear.

Given the prolonged and repeated periods of lockdown, it is important to evaluate the overall effect on childhood immunisation uptake. The aim of this study was, therefore, to provide a longer perspective than previous studies by describing the pattern of childhood immunisation uptake in Scotland and England before, during, and immediately after the first national lockdown implemented in response to the pandemic (23 March to 31 July), with comparisons to baseline data from 2019, by geographical area and socioeconomic deprivation.

## Methods

### Study design

This observational study took advantage of the natural experiment afforded by the COVID-19 pandemic and used routinely collected data for the year prior to the pandemic (2019) and immediately before, during, and after the first period of "lockdown" imposed by the United Kingdom and Scottish governments in 2020. Data were available for Scotland and England; however, as discussed below, variations in time points at which the data were collected precluded direct comparisons. Of note, this analysis relates to the first national lockdown, which began on 23 March 2020, with restrictions easing gradually from over June and July 2020. The prespecified analysis plan is included within the Supporting information (S1 File). Minor changes were made to the planned visualisation format of the data (from bar to line plots) to better illustrate the findings.

The vaccines chosen as indicators of preschool immunisation uptake were the hexavalent DTaP/IPV/Hib/HepB vaccine, (referred to here as "6-in-1"), which protects against diphtheria, tetanus, pertussis, polio, *Haemophilus influenzae* type b, and hepatitis B, and the measles, mumps, and rubella vaccine (referred to as "MMR"). In the UK, the 6-in-1 vaccine is recommended at age 8 ("first dose 6-in-1"), 12 ("second dose 6-in-1"), and 16 ("third dose 6-in-1") weeks of age, and MMR is given at 12 months ("first dose MMR") and 3 years 4 months ("second dose MMR") [16]. Uptake of the additional immunisations offered at the same ages (Meningococcal C, Rotavirus, and Pneumococcal) were not assessed.

For Scotland, we chose to primarily examine uptake within 4 weeks of eligibility as this represents timely uptake of vaccinations as per the recommended UK schedule [16] leading to the

child being protected at the earliest recommended opportunity. All children living in Scotland who became eligible by age for any of the preschool immunisations of interest from January 2019 up to and including the week beginning 28 September 2020 were included. Of secondary interest, and to allow descriptive comparisons with data from England, we also analysed uptake at approximately 6 months (range 24 to 32 weeks) for the 6-in-1 and 16 months for the first dose MMR. For England, equivalent data on uptake within 4 weeks were not available; therefore, the analysis was conducted on uptake by six (6-in-1) or 18 months of age (first MMR), and monthly, rather than weekly, data were used.

Vaccine uptake was analysed in the following 4 time periods: 1 January to 31 December 2019 ("2019"), 1 January to week beginning 16 March 2020 ("prelockdown"), 23 March to week beginning 27 July ("lockdown") and week beginning 3 August to week beginning 28 September 2020 inclusive ("postlockdown"). These time periods were chosen to correspond with the beginning of the first UK-wide lockdown as announced by the UK government on 23 March 2020 [1]. The end of the lockdown period is less well defined and varied both in approach and timescale between Scotland and England [17,18]. Broadly speaking, by the end of July 2020, there was a substantial reduction in "lockdown" restrictions in both countries with the opening of many nonessential businesses and limited indoor meeting between households permitted; therefore, a pragmatic approach was taken to define the lockdown period as 23 March 2020 until 31 July 2020. Children were included in the time period at which they first became eligible by age. As data from England were available by month only, the prelockdown period included January to end of March 2020.

## Data sources

The Scottish data used in this paper were extracted in January 2021 from the PHS "COVID19 wider impacts on the health care system" dashboard, which is publicly accessible at https://scotland.shinyapps.io/phs-covid-wider-impact/. The dashboard includes aggregate information on immunisation uptake, including by the geographical area in which the child is living and the corresponding Scottish Index of Multiple Deprivation (SIMD) (both assigned based on the child's postcode registered on the Scottish national vaccination call-recall system, SIRS). The SIMD breaks Scotland into 6,976 small areas of similar population size and assigns one of 5 SIMD quintiles based on indicators of deprivation including income, education, and housing, with 1 representing the most deprived areas and 5, the least [19]. We defined geographical areas by the Health and Social Care Partnership (HSCP) in which the child lives. Within Scotland, there are 31 HSCPs that provide integrated health and social care to their population. Additional information on inclusion and exclusion criteria can be found in S2 File.

English data were extracted from the ImmForm system, for January 2019 (representing the first extract for prelockdown period) to September 2020 (representing the final extract for postlockdown period), which automatically receives monthly aggregate data on vaccine uptake from 92% to 95% of English GP practices, provided by GP System Suppliers. These data are validated and analysed by Public Health England to check completeness, query any anomalous results, and are used to describe epidemiological trends, as well as being used directly locally by the NHS for performance management. Data were available for immunisation uptake at age 6 months (each dose of the 6-in-1 immunisation) and 18 months (first MMR) only.

## Statistical methods

The primary outcome examined was the change in percentage uptake, within 4 weeks of eligibility, of each of the immunisations of interest between baseline uptake rates in 2019 and during the lockdown period for Scotland as a whole and by geographical areas or level of

deprivation. Of secondary interest were comparisons between 2019 and the other time points listed above. Broadly comparable analyses were conducted to examine the primary outcome for England as a whole. Due to differences in data collection methods and age of child at data extraction point, comparisons between Scottish and English data were descriptive only.

To compare uptake rates between time periods, aggregate binary logistic regression modelling was conducted, using vaccination status (vaccinated versus unvaccinated) as the dependent variable, and time period as the explanatory variable. Separate analyses were carried out for each vaccine and country. When comparing HSCP or deprivation, an additional interaction with HSPC or SIMD quintile was included in the model. Odds ratios (ORs) with 95% confidence intervals (CIs) were calculated for uptake in the period of interest compared to the 2019 baseline. Given the nature of the aggregated data available, no adjusting for potential confounders or effect modifiers was possible. Further methodological details can be found in S2 File.

All analyses and generation of figures were performed using R/R Studio (4.0.3). All code is publicly available via the EAVE II GitHub page (https://github.com/EAVE-II). Scottish data can be accessed via the PHS wider impacts dashboard https://scotland.shinyapps.io/phs-covid-wider-impact/. The English data used are included in the Supporting information, and any further queries should be directed to EarlyChildhoodBaseline@phe.gov.uk.

### Ethics and funding

Ethical approval for this specific study was not required as we have used publicly available, anonymised, aggregated data. This study is reported as per the REporting of studies Conducted using Observational Routinely collected Data (RECORD) guideline (S3 File) [20]. No specific funding was received for this project; RM is funded by the Health Data Research UK BREATHE hub.

## Results

Data were available for the outcome of 439,754 vaccine invitations in Scotland and 4.08 million for England across all time periods. As many infants or children would be eligible for multiple immunisations during the study, these figures do not represent individual participant numbers. Detailed demographic data are not routinely collected and were therefore not available. For context, S1 Table shows population data from the relevant national records offices for births in Scotland and England by sex, maternal age, and maternal country of birth from 2015 to 2020. It is expected that these data are broadly representative of the children included in this study.

All CIs presented are 95% CIs, and ORs are compared to the baseline year 2019 unless otherwise stated.

### Preschool immunisation uptake increased during the lockdown period in Scotland

Across Scotland, the percentage of preschool children receiving their immunisations within 4 weeks of becoming eligible increased during the lockdown period for all 5 immunisations (Fig 1, Table 1, S2 Table). The change in percentage uptake compared to the 2019 baseline ranged from 1.3% for the first dose 6-in-1 (OR 1.28, CI 1.18 to 1.39) to 14.3% for the second dose MMR (OR 1.8, CI 1.74 to 1.87) (Table 1). Across all the immunisations visits, this equated to an additional 7,508 preschool immunisations being delivered in a timely manner over the lockdown period compared to the baseline rates of 2019. Uptake rates dipped immediately before the announcement of a national lockdown in mid-March 2020 (Fig 1), then peaked

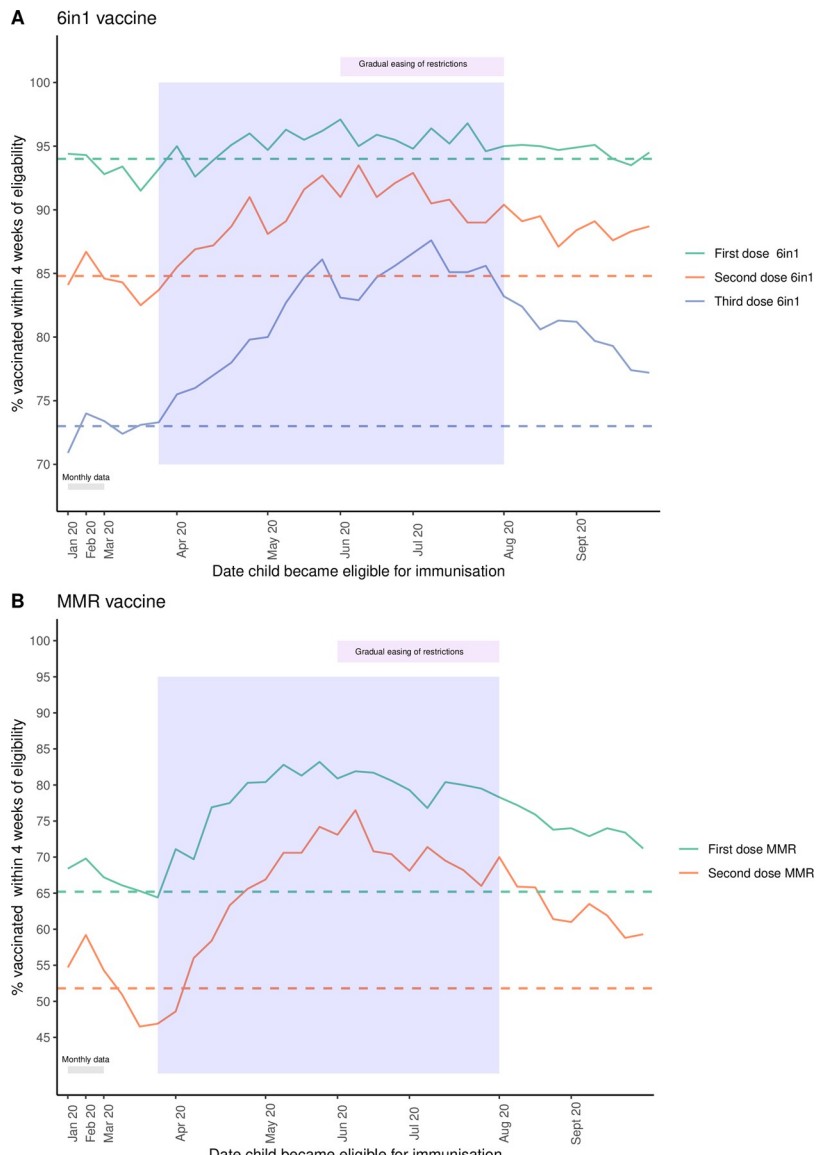

**Fig 1. Percentage of children in Scotland immunised within 4 weeks of eligibility.** (A) All doses 6-in-1 vaccine; (B) MMR vaccine across January to September 2020 in Scotland. Lockdown period = blue shaded area. For January and February, a single mean monthly value is plotted, and from March onwards, weekly uptake is shown (see S2 Table for full data). Dashed horizontal lines (—) indicated the mean uptake in 2019 of the immunisation with the corresponding colour. The increase in uptake during the lockdown period was statistically significant (see Table 1 for details). MMR, measles, mumps, and rubella.

throughout June before starting to decrease. However, uptake remained significantly higher than 2019 during the postlockdown period (Fig 1, Table 1). Of note, prior to lockdown for both MMR doses, there was already a modest, increase in uptake compared to 2019 (Table 1).

## Variation in uptake of preschool immunisations by geographical area

Baseline data from 2019 showed the percentage uptake of preschool immunisations within 4 weeks of eligibility varied widely by geographical HSCP and immunisation (Fig 2, S1 Fig, S3 Table). In keeping with the rise in mean uptake across Scotland for all vaccines, the percentage

**Table 1. Uptake of preschool immunisations in Scotland by time period with OR compared to baseline of 2019.**

| Immunisation | Time period | % uptake within 4 weeks of eligibility (number received/total eligible) | % point change from 2019 | OR for uptake compared to 2019 (95% CI) | p-Value |
|---|---|---|---|---|---|
| First dose 6-in-1 | 2019 | 94 (47,567/50,609) | NA | NA | NA |
| | Pre LD | 93.3 (10,103/10,761) | −0.7 | 0.98 (0.9–1.07) | 0.68 |
| | LD | 95.3 (16,371/17,133) | 1.3 | 1.28 (1.18–1.39) | <0.001 |
| | Post LD | 94.6 (8,075/8,531) | 0.6 | 1.13 (1.02–1.25) | 0.02 |
| Second dose 6-in-1 | 2019 | 84.8 (43,221/50,975) | 0 | NA | NA |
| | Pre LD | 84.4 (9,106/10,698) | −0.4 | 1.03 (0.97–1.09) | 0.39 |
| | LD | 89.7 (15,443/17,222) | 4.9 | 1.56 (1.47–1.65) | <0.001 |
| | Post LD | 88.7 (7,459/8,412) | 3.9 | 1.4 (1.31–1.51) | <0.001 |
| Third dose 6-in-1 | 2019 | 73 (37,266/51,083) | NA | NA | NA |
| | Pre LD | 72.8 (8,276/11,394) | −0.2 | 0.98 (0.94–1.03) | 0.49 |
| | LD | 82.1 (14,039/17,093) | 9.1 | 1.7 (1.63–1.78) | <0.001 |
| | Post LD | 80.3 (6,555/8,172) | 7.3 | 1.5 (1.42–1.59) | <0.001 |
| First dose MMR | 2019 | 65.2 (33,935/52,015) | NA | NA | NA |
| | Pre LD | 67.4 (7,782/11,370) | 2.2 | 1.16 (1.11–1.21) | <0.001 |
| | LD | 78.4 (14,482/18,463) | 13.2 | 1.94 (1.86–2.02) | <0.001 |
| | Post LD | 74.5 (6,740/9,047) | 9.3 | 1.56 (1.48–1.64) | <0.001 |
| Second dose MMR | 2019 | 51.8 (25,844/49,940) | NA | NA | NA |
| | Pre LD | 53.1 (6,390/11,495) | 1.3 | 1.17 (1.12–1.22) | <0.001 |
| | LD | 66.1 (11,303/17,145) | 14.3 | 1.8 (1.74–1.87) | <0.001 |
| | Post LD | 63.1 (5,171/8,196) | 11.3 | 1.59 (1.52–1.67) | <0.001 |

p-Value calculated using aggregate binary logistic regression, statistically significant changes in uptake compared to 2019 are shaded green. p-Value rounded to 2 decimal places.

CI, confidence interval; LD, lockdown; MMR, measles, mumps, and rubella; NA, not applicable; OR, odds ratio; Post LD, postlockdown; Pre LD, prelockdown.

of children immunised in most HSCP increased between 2019 and lockdown. However, not all followed this pattern with a minority demonstrating a fall in uptake (Fig 2, S1 Fig, S3 Table). Care must be taken when interpreting percentage results from the island HSCPs (Shetland Islands, Orkney Islands, and Western Isles) given the very small numbers of children involved (S3 Table).

For individual HSCP, the statistical significance of the change in uptake varied by immunisation (Fig 2, S3 Table). Despite a general trend of improvement for the first 6-in-1 vaccine, we found a significant change for only 8 of the 31 HSCP, mainly centred around the more densely populated, urban central belt of Scotland (Glasgow City, Edinburgh, Stirling and Clackmannanshire, East Dunbartonshire, Falkirk, Fife, South Lanarkshire, South Ayrshire; Fig 2). However, this pattern evolved with the different immunisation visits and with almost all HSCP showing a rise in uptake for both MMR immunisations, with percentage point increases as high as 30% (Angus, 74.1% versus 43.8%, OR 3.38,CI 2.61 to 4.37) (Fig 2, S3 Table).

## Preschool immunisation uptake increased across all deprivation levels

Percentage uptake within 4 weeks of becoming eligible rose across all SIMD quintiles, between 2019 and lockdown, for all immunisations (Fig 3). The magnitude of this rise varied by quintile and vaccine (S2 Fig, S4 Table) from 0.3% (SIMD 4, first 6-in-1 dose, OR 1.1, 95% CI 0.9 to 1.3) to 16.2% (SIMD 5, second MMR dose, OR 2.1, 95% CI 1.9 to 2.3). The increase in uptake between 2019 and lockdown was statistically significant for all except first dose 6-in-1 for the

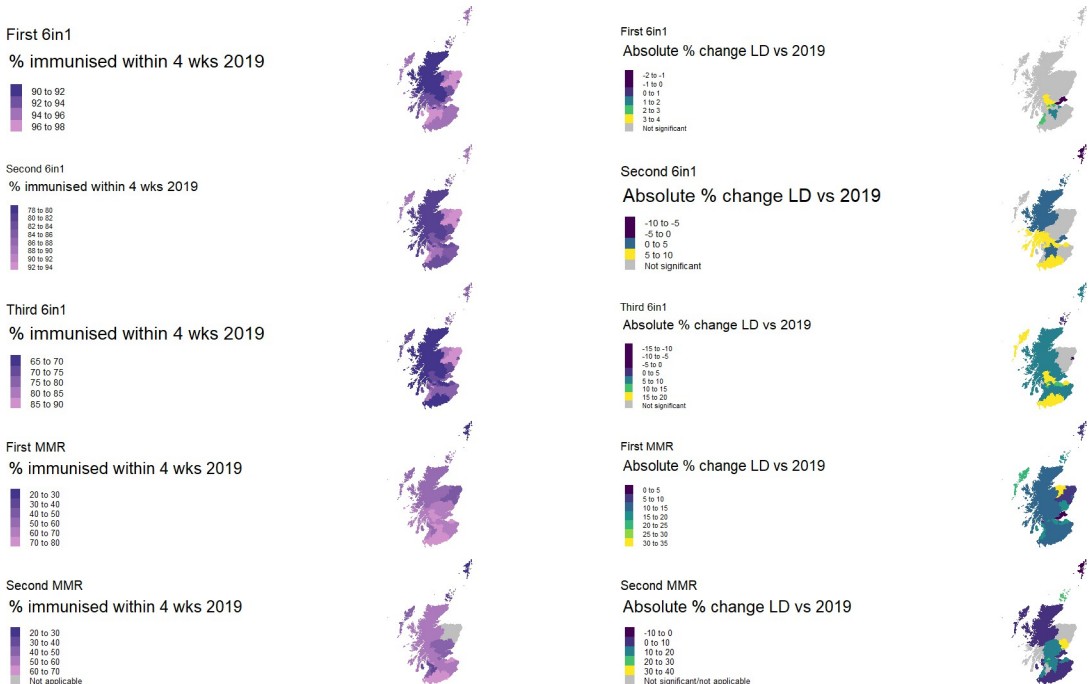

**Fig 2. Percentage uptake by Scottish HSCP in 2019 and statistically significant increases between 2019 and "lockdown".**
Left-hand side maps show baseline mean percentage uptake by HSCP in Scotland in 2019, and right-hand side maps show percentage point difference between 2019 and lockdown for areas in which the difference was statistically significant for each vaccine. Grey areas indicated HSCP where there were no statistically significant changes in uptake between 2019 and lockdown. Note that in Grampian, the second MMR dose is delivered at a later age; therefore, data for this area are recorded as "nonapplicable" (further details in S2 File). The full dataset can be found in S3 Table. Base layer map available from https://spatialdata.gov.scot/geonetwork/srv/eng/catalog.search;jsessionid=714BD98E15D22A8116824CA25B30DC02#/metadata/ac5e870f-8fc2-4c21-8b9c-3bd2311a583f, which contains public sector information licenced under the Open Government Licence v3.0. HSCP, Health and Social Care Partnership; LD, lockdown; MMR, measles, mumps, and rubella; Wks, weeks.

least deprived quintiles (4 and 5) (S4 Table). In the postlockdown period, percentage uptake remained significantly higher than the 2019 baseline for all quintiles for each vaccine except for the first 6-in-1 dose, in which only the most deprived quintile retained a significant increase (S2 Fig, S4 Table).

In keeping with previous observations [21], children in the least deprived quintiles were more likely to be immunised and this relationship was broadly maintained throughout the study period (Fig 3 and S3). While all quintiles improved uptake between 2019 and lockdown, whether the inequality between most and least deprived increased or decreased varied by vaccine type. For all doses of the 6-in-1 vaccine, there was a tendency to a convergent improvement, i.e., the gap in percentage uptake between the quintiles lessened, while for both MMR doses, there was further divergence in uptake rates between most and least deprived (Fig 3, S4 Fig, S5 Table). The interaction of SIMD quintile and time period was nonsignificant for all 6-in-1 doses; that is to say all SIMD quintiles improved equally for this immunisation (S5 Table). However, for the first MMR dose, the improvement in uptake was statistically greater for SIMD quintiles 3 to 5 compared to SIMD 1, and for the second MMR dose, SIMD 5 showed a significantly larger increase in uptake between 2019 and lockdown (S5 Table). This suggests that for the MMR immunisation, the factors leading to an improvement in uptake had greater positive impact for children living in less deprived areas.

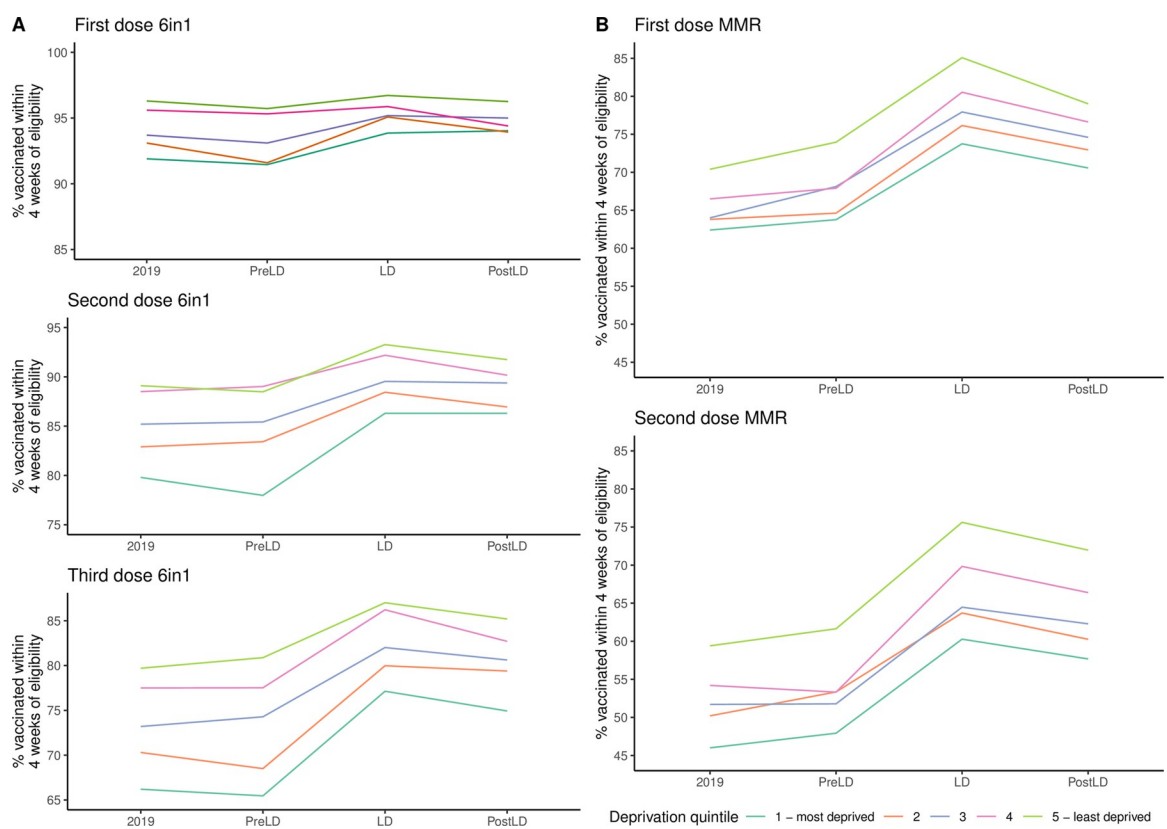

**Fig 3. Mean percentage immunised by SIMD quintile in Scotland.** (A) 6-in-1 vaccine; (B) MMR vaccine for Scotland. See S2 Fig and S4 Table for absolute percentage rise compared to 2019 and significance levels. LD, lockdown; MMR, measles, mumps, and rubella; PostLD, postlockdown; PreLD, prelockdown; SIMD, Scottish Index of Multiple Deprivation.

### "Catch-up" immunisation rates and comparison with data from England

Thus far, the Scottish data presented have related to children receiving their immunisations as per the recommended schedule (within 4 weeks of the child becoming eligible by age), representing a "gold standard" in which the child is protected as early as possible. It is recognised that some children will receive their immunisations after this point. This "catch-up" effect can be seen in uptakes rate for all Scottish children by the time they reached between 6 and 8 months (6-in-1), 16 months (first MMR), or 3 years 8 months (second MMR) (Fig 4A). Those becoming eligible during lockdown showed minimal change in this longer-term measure of uptake for the 3 doses of the 6-in-1 immunisation, a small increase in uptake of the first MMR, and a more substantial increase in uptake of the second MMR (S6 Table). These data suggest that while lockdown was associated with a beneficial effect on timely uptake of all infant and preschool immunisations, the impact on longer term or final achieved uptake was more variable, possibly reflecting a ceiling effect on maximal uptake, for the earliest immunisations.

For England, broadly equivalent data were available for children aged 6 months (all doses 6-in-1) and 18 months (first dose MMR) who had become eligible for their immunisations during the time periods of interest (Fig 4B, S7 Table). These data demonstrated a small but statistically significant fall in uptake for all the immunisations in the lockdown periods compared to 2019, ranging from 0.5% (first dose 6-in-1, 95.8% versus 96.3%, OR 0.89, CI 0.86 to 0.91) to 2.1% (third dose 6-in-1, 86.6% versus 88.7, OR 0.82, CI 0.81 to 0.83) (S7 Table). However, much of the fall in uptake took place in the prelockdown period, particularly for the third dose

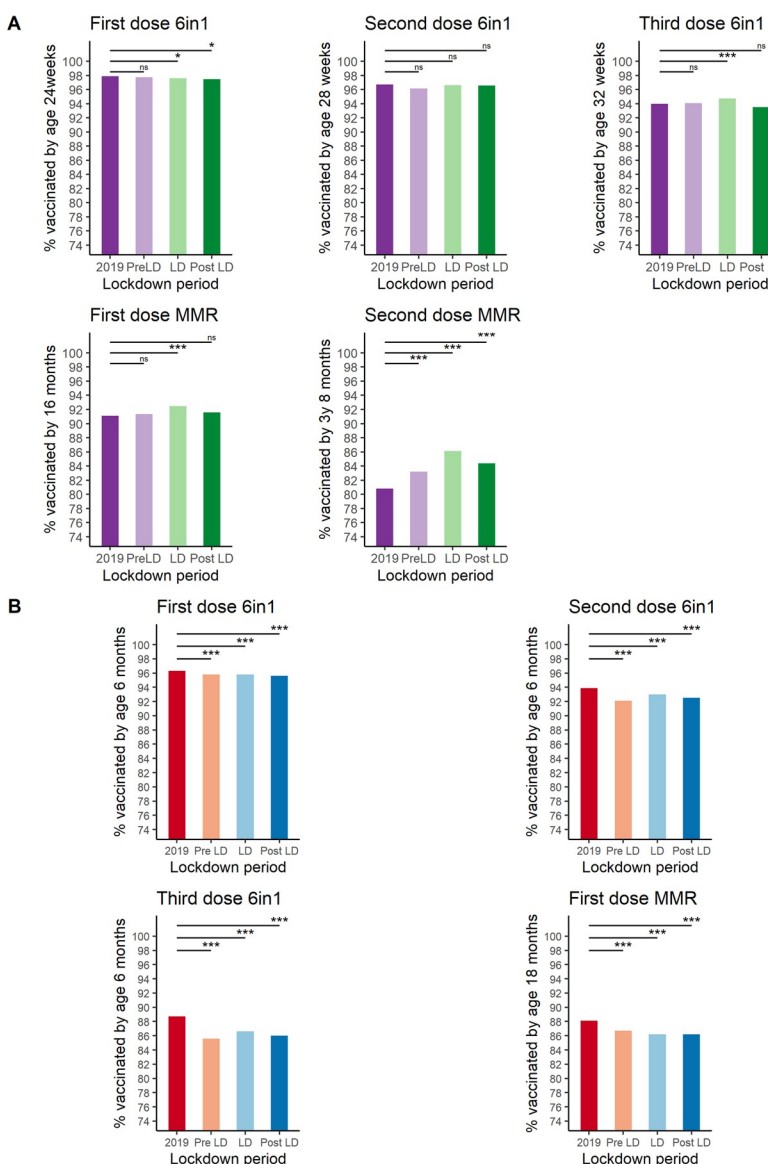

**Fig 4. Percentage of children up to date at 6/18 months in Scotland and England.** Overall mean percentage of children immunised by approximately 6 months of age (all doses of 6 in1, see y-axis for specific ages) or 16–18 months (first MMR) for Scotland and England. Each bar contains children who became eligible for the immunisation of interest during the time period indicated. (A) Scotland, (B) England *** *p*-value < 0.001, * *p*-value <0.05, ns, not significant. *p*-values calculated using binary logistic regression. LD, lockdown; MMR, measles, mumps, and rubella; PostLD, postlockdown; PreLD, prelockdown.

6-in-1, with a gradual recovery seen over the lockdown period itself (S4 Fig). A general trend towards falling MMR uptake can also be seen to predate the COVID-19 pandemic (S4 Fig).

## Discussion

We found that early uptake of infant and preschool immunisations (within 4 weeks of a child becoming eligible) rose significantly for the duration of the first lockdown period in Scotland, resulting in thousands more children receiving their immunisations at the scheduled time. Encouragingly, positive results were seen across all levels of deprivation, though some

geographical variations were observed across Scotland. Improvement was also seen in longer term uptake of the first and second MMR immunisations (immunisation within 4 months of becoming eligible) in the lockdown period. Findings in England differed, with a small fall in longer-term uptake of immunisations observed for the lockdown period.

The Scottish data in particular challenge the early predictions of an adverse effect of lockdown measures on childhood immunisation uptake and demonstrate the importance of continued surveillance throughout the various phases of pandemic control measures. From these data alone, we are unable to fully determine what changes in both the barriers and motivators to vaccination may have led to this result; however, examining previous research can offer some suggestions. While the concept of vaccine hesitancy is a popular media topic, previous studies have shown that the reasons given by parents for not vaccinating their children are often much more practical. A prepandemic report by the Royal Society for Public Health in the UK found that the major barriers for parents who wanted to immunise their children were timing and availability of appointments and childcare duties [22]. For those who actively chose not to immunise their children, fear of side effects was a key concern and the negative effects of social media are increasing [22]. The lockdown has had a major effect on parental working patterns with almost 9 million UK employees furloughed and millions more working from home, often with the additional tasks of managing childcare as schools and nurseries were closed [23]. While this had made life significantly more challenging for many, more flexible working patterns may have made attending immunisation appointments easier for some.

Jarchow-MacDonald and colleagues [24] from NHS Lothian (which consists of HSCPs Edinburgh, West Lothian, East Lothian, and Midlothian) have suggested that ensuring the accessibility of immunisation centres, either by public transport or by providing mobile services to shielding families, was important in maintaining uptake during the pandemic, as was directly communicating with families with a preappointment phone call and reminder postcards. This gave families an opportunity to discuss the immunisation with a healthcare professional, a strategy that has been showed to be important in addressing parental concerns [25]. In fact, the reminder alone may have been sufficient to encourage parents to attend the first appointment [26]. The clear commitment of the Scottish Government to maintain the immunisation programme was also felt to be an important factor [24].

In England, Bell and colleagues [27] conducted a large-scale online survey of parents of children under the age of 18 months to assess their experiences of accessing immunisations during the early part of the UK lockdown (19 April to 11 May 2020). They highlighted the uncertainty of some families about whether the immunisation service was continuing, particularly among non-white ethnic groups. While most felt it was important that their child was immunised on time, Bell and colleagues reported that many parents felt the risk of catching a vaccine-preventable infectious disease was reduced due to limited mixing with others. This suggests that an enhanced appreciation of the utility of immunisation was not a major motivator for parents to ensure they attended the immunisation appointments. These issues may well be reflected in the fall in immunisation uptake in England during the earlier part of lockdown reported here, and a delay in receiving the first dose 6-in-1 may well led to delays in subsequent doses, meaning that the infants were unable to "catch up" by 6 months of age.

Other factors, which may have had an impact on promoting timely immunisation uptake, could include a reduction in fever, cough, and colds, which may otherwise have caused parents to delay immunisations. Though specific data on this point are lacking, a significant fall in the detection of Rhinovirus in adults was observed in 2020 compared to 2019 [28], suggesting that "normal" childhood respiratory infections are likely to have decreased. More work is required to fully dissect the key factors in improving timely preschool immunisation uptake in Scotland. Clarity and publicity about the continuation of the immunisation programmes, telephone

reminders, and the opportunity to discuss with healthcare professions seem likely to have had the most impact. We plan to use the data generated in this study to inform future work investigating these factors, including geographical differences and the introduction of policies such as reminders and publicity campaigns.

It is important to acknowledge the limitations of the data presented, many of which arise from opportunistically using routinely collected data rather than that obtained from a specific study design. The SIRS electronic system is well established and captures data on the entire child population in Scotland. However, the aggregate surveillance data derived from the system that we could access lacked detailed information on several potentially important factors, not least of which was ethnicity, which is known to affect both immunisation uptake and attitudes towards immunisation. [29]. As shown in S1 Table, the proportion of mothers born outside the UK is relatively low in Scotland (16.3% to 17.7% between 2015 and 2020) and is almost half that of England (28.4% to 30.2%). It may not be appropriate to extrapolate these data to countries with a significantly different ethnic makeup, and it is plausible that some of the difference seen between the Scottish and English data could be due to these factors.

In using the mean percentage of the entire year 2019 as our baseline comparator, we potentially run the risk of confusing normal seasonal variation in immunisation uptake with the impact of lockdown measures. Ideally, direct weekly or monthly comparisons would be made between 2019 and 2020, though these data are not available. The enhanced detail and frequency of collection of immunisation uptake data and rapid dissemination (through the COVID dashboard) was in response to the pandemic and therefore only began in March 2020. However, quarterly trends published for previous years including 2019 do not show major difference in uptake throughout the year and in fact show a gradual decline in uptake year on year since 2015 [30]. In addition, it is possible that 2020 uptake rates have been underestimated, due to a lag in data entry into SIRS, which would particularly affect the "catch-up" rates (Fig 4A).

Therefore, caution must be taken not to overinterpret the results presented here or extrapolate to significantly different populations with varying baseline immunisation uptakes rates and less robust immunisation programmes, the organisation of which may have been adversely affected by the pandemic. Nevertheless, this study has efficiently and quickly produced useful and valid results, which have the potential to aid the development of future research and guide policy. The positive findings regarding immunisation uptake despite the extreme circumstances of the COVID-19 pandemic and associated lockdowns (in Scotland at least) is an important message to send to support public and professional confidence in the preschool immunisation programme and help normalise timely immunisation uptake for both parents and health services. Improving public and professional confidence is particularly vital given the current importance of promoting vaccination against SARS-CoV-2. Despite these encouraging data, it is not possible to ascertain from the numbers alone, which are the key contributing factors to improving uptake. This is a key avenue of future research as lessons learnt can then be taken forward to optimise future immunisation programmes, both within the pandemic setting and beyond. In addition, we have not yet assessed the potential impact of improved early immunisation uptake on rates of vaccine preventable diseases. Any observed changes in vaccine-preventable diseases (for example, pertussis in young babies) would be heavily confounded by the changes in behaviour such as social distancing and isolation for symptoms such as cough. However, of the 5 vaccinations, there did appear to be an increase in overall coverage for the second MMR; therefore, monitoring trends in measles cases as we move fully out of COVID-related restrictions may be of interest, though we would not be able to infer causation.

The ongoing COVID-19 pandemic continues to stretch health services and adversely affect all areas of life, with children disproportionately bearing the burden of the indirect consequences of pandemic control measures such as the closure of schools and limited social contact. However, opportunities have also been created in terms of enhanced surveillance of health programmes. In this study, we have used such data to investigate the effect of the pandemic on infant and preschool immunisation uptake. We have demonstrated that a robust child immunisation service can continue to deliver high and even increasing uptake rates. Families will respond despite the many difficulties they face, to ensure that children continue to be protected again vaccine-preventable diseases. The challenge now is to use and expand on this knowledge to promote future vaccination programmes, including those targeting SARS-CoV-2.

## Supporting information

**S1 Fig.** Percentage uptake by Scottish HSCP for 2019 (pale orange) and LD (dark orange) with HSCP ordered by uptake for 2019 (note: this varies by immunisation). Dashed horizontal lines indicated the mean uptake for all of Scotland for the time period of the corresponding colour. HSCP, Health and Social Care Partnership; LD, lockdown.
(TIF)

**S2 Fig.** Absolute percentage change in uptake in Scotland compared to 2019 for each immunisations and SIMD for each time period (A = Pre LD, B = LD, C = Post LD). Significance rates varied by immunisation and SIMD; for details, see S3 Table. LD, lockdown; Post LD, postlockdown; Pre LD, prelockdown; SIMD, Scottish Index of Multiple Deprivation.
(TIF)

**S3 Fig.** Combined OR plot with 95% CI comparing each SIMD quintile (2–5) to SIMD 1-most deprived for 2019 (dark blue) and LD (light blue). Data for Scotland only. CI, confidence interval; LD, lockdown; OR, odds ratio; SIMD, Scottish Index of Multiple Deprivation.
(TIF)

**S4 Fig. Percentage of children in England immunised by 6 months of age (first, second, and third dose 6-in-1) or 18 months of age (first dose MMR) from January 2019 to September 2020.** The start and end of the lockdown period is indicated by the purple shaded area. MMR, measles, mumps, and rubella.
(TIF)

**S1 Table. Summary statistics for all live births in Scotland and England 2015–2020 from the National Records of Scotland (Scotland) and Office of National Statistics (England).** Note this table gives an overview of the population as these data were not available for the individuals within the study. The date range encompasses the oldest to youngest children potentially included. For example, those born in 2015 would be eligible for the second dose MMR in 2019 and those born in early August 2020 would be eligible for the first dose 6-in-1 during the "postlockdown" time period. However, this does not account for children who may have migrated into/out of the areas since birth. Percentages are rounded to 1 decimal place. MMR, measles, mumps, and rubella.
(DOCX)

**S2 Table. Percentage uptake of each immunisation for Scotland by year (2019), month (January and February 2020), or week as per data availability.** W/B, week beginning.
(DOCX)

**S3 Table.  A-E:** Percentage uptake, percent point change in uptake compared to 2019 and significance level for this change for each HSCP at each time period. Each table shows results for a different immunisation. *p*-Values calculated using aggregate binary logistic regression and rounded to 2 decimal places. Results were considered significant if *p*-value <0.05 and 95% CI did not include 1. Statistically significant *p*-values are shaded green, and significant results for the 2019–LD comparisons are plotted on Fig 2. CI, confidence interval; HSCP, Health and Social Care Partnership; LD, lockdown; NA, not applicable; OR, odds ratio; PostLD, postlockdown; PreLD, pre lockdown.
(DOCX)

**S4 Table. Uptake of preschool immunisations by time period and SIMD and percent point change in uptake compared to baseline 2019.** OR and 95% CI shown are for change in uptake compared to 2019. *p*-Values calculated using aggregate binary logistic regression and rounded to 2 decimal places. Statistically significant change in uptake compared to 2019 are shaded green. CI, confidence interval; LD, lockdown, NA, not applicable; OR, odds ratio; SIMD, Scottish Index of Multiple Deprivation.
(DOCX)

**S5 Table. To assess whether the differences between change in uptake were statistically significant between SIMD quintiles, the interaction between time period and SIMD quintile was added into the model.** The baseline comparisons showed are for time period 2019 and deprivation quintile SIMD 1. If the 95% CI did not include 1, the interaction of time period and SIMD was considered statistically significant, that is; there was a significant difference in the level of change (2019- time period) between the deprivation quintile and SIMD 1. For example, the increase in uptake during lockdown for SIMD 5 was statistically greater than the increase in uptake for SIMD 1. The ROR can be used to calculate the OR for uptake compared to the baseline levels by multiplying the ROR with the relevant OR in S3 table. *p*-Values calculated using aggregate binary logistic regression and rounded to 2 decimal places. CI, confidence interval; LD, lockdown; ns, not statistically significant (coloured green), interaction was statistically significant; OR, odds ratio; ROR, ratio of odds ratio, calculated by taking the exponential function of the coefficient of the interaction term from the interaction model; SIMD, Scottish Index of Multiple Deprivation.
(DOCX)

**S6 Table. Scotland.** Uptake of preschool immunisations at an older age by time period and point percentage change from 2019 with OR and 95% CI compared to baseline of 2019. Children are categorised into the time period at which they became eligible for the immunisation as before and uptake data were extracted at a later stage when they reached the ages indicated in the immunisation column. Statistically significant changes are coloured green. *p*-Values calculated using aggregate binary logistic regression and rounded to 2 decimal places. CI, confidence interval; LD, lockdown; NA, not applicable; OR, odds ratio.
(DOCX)

**S7 Table. England.** Uptake of preschool immunisations at an older age by time period and point percentage change from 2019 with OR and 95% CI compared to baseline of 2019. Children are categorised into the time period at which they became eligible for the immunisation as before and uptake data were extracted at a later stage when they reached the ages indicated in the immunisation column. Statistically significant changes are coloured green. *p*-Values calculated using aggregate binary logistic regression and rounded to 2 decimal places. CI, confidence interval; LD, lockdown; NA, not applicable; OR, odds ratio.
(DOCX)

**S1 File. Childhood immunisation V1.0 final analysis plan.**
(DOCX)

**S2 File. Supplemental methods.**
(DOCX)

**S3 File. RECORD plus STROBE checklist.**
(DOCX)

## Acknowledgments

We thank Public Health Scotland for making the data publicly available and acknowledge the support of the HDR UK BREATHE Hub and EAVE II collaborators.

## Author Contributions

**Conceptualization:** Fiona McQuaid, Rachael Wood.

**Data curation:** Fiona McQuaid, Rachel Mulholland, Yuma Sangpang Rai, Judith Tait, Jaime Villacampa Ortega.

**Formal analysis:** Fiona McQuaid, Rachel Mulholland, Yuma Sangpang Rai, Elise Tessier.

**Investigation:** Fiona McQuaid.

**Methodology:** Fiona McQuaid, Rachel Mulholland, Helen Bedford, J. Claire Cameron, Cheryl Gibbons, Aziz Sheikh, Judith Tait, Steve Turner, Jaime Villacampa Ortega, Rachael Wood.

**Project administration:** Fiona McQuaid.

**Resources:** Fiona McQuaid, Judith Tait, Jaime Villacampa Ortega.

**Software:** Fiona McQuaid, Rachel Mulholland.

**Supervision:** Partho Roy, Elise Tessier, Joanne White, Rachael Wood.

**Validation:** Fiona McQuaid.

**Visualization:** Fiona McQuaid.

**Writing – original draft:** Fiona McQuaid.

**Writing – review & editing:** Rachel Mulholland, Yuma Sangpang Rai, Utkarsh Agrawal, Helen Bedford, J. Claire Cameron, Cheryl Gibbons, Partho Roy, Aziz Sheikh, Ting Shi, Colin R. Simpson, Judith Tait, Elise Tessier, Steve Turner, Jaime Villacampa Ortega, Joanne White, Rachael Wood.

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
