## [Editor Report · Decision Letter 0]

23 Jul 2021

Dear Dr McQuaid, 

Thank you for submitting your manuscript entitled "Uptake of infant and pre-school immunisations in Scotland and England during the COVID-19 pandemic: an observational study of routinely collected data" for consideration by PLOS Medicine.

Your manuscript has now been evaluated by the PLOS Medicine editorial staff and I am writing to let you know that we would like to send your submission out for external peer review.

Kind regards,

Louise Gaynor-Brook, MBBS PhD

Associate Editor

PLOS Medicine

---

## [Decision Letter · Decision Letter 1]

29 Sep 2021

Dear Dr. McQuaid,

Thank you very much for submitting your manuscript "Uptake of infant and pre-school immunisations in Scotland and England during the COVID-19 pandemic: an observational study of routinely collected data" (PMEDICINE-D-21-03163R1) for consideration at PLOS Medicine. 

Your paper was evaluated by three independent reviewers, including a statistical reviewer, and was discussed among all the editors here and with an academic editor with relevant expertise. The reviews are appended at the bottom of this email and any accompanying reviewer attachments can be seen via the link below:

[LINK]

In light of these reviews, I am afraid that we will not be able to accept the manuscript for publication in the journal in its current form, but we would like to consider a revised version that addresses the reviewers' and editors' comments. Obviously we cannot make any decision about publication until we have seen the revised manuscript and your response, and we plan to seek re-review by one or more of the reviewers. 

We expect to receive your revised manuscript by Oct 20 2021 11:59PM. Please email us (plosmedicine@plos.org) if you have any questions or concerns.

We look forward to receiving your revised manuscript. 

Sincerely,

Louise Gaynor-Brook, MBBS PhD

Associate Editor, PLOS Medicine

plosmedicine.org

Comments from the Academic Editor: 

As the reviewers indicate, there is a need of data and descriptions of the secondary effects of the pandemic on health services, coverage and health outcomes. A possible increase in coverage is of interest among all other reports of negative indirect effects - although it may be difficult to draw inferences that are relevant to other settings.

General comments:

Throughout the paper, please adapt reference call-outs to the following style: "... likely to be profound [7,8]." (noting the absence of spaces within the square brackets).

Data availability:

PLOS Medicine requires that the de-identified data underlying the specific results in a published article be made available, without restrictions on access, in a public repository or as Supporting Information at the time of article publication, provided it is legal and ethical to do so. 

Where the data are not freely available, please describe briefly the ethical, legal, or contractual restriction that prevents you from sharing it. Please also include an appropriate contact (web or email address) for inquiries - please note that a study author cannot be the contact person for data requests.

Abstract Background: The final sentence should clearly state the study question.

Abstract Methods and Findings:

Please provide brief demographic details of the study population (e.g. sex, age, ethnicity, etc)

Please include the number of participants and dates between which the study took place. 

Please define abbreviations at first use e.g. DTaP/IPV/Hib/HepB, MMR

Please clarify whether CIs presented are 95% CIs.

Where ORs are provided, please specify the comparison group.

In the last sentence of the Abstract Methods and Findings section, please describe 2-3 of the main limitations of the study's methodology."

Abstract Conclusions:

Please begin your Abstract Conclusions with "In this study, we observed ..." or similar, to summarize the main findings from your study, without overstating your conclusions. Please emphasize what is new and address the implications of your study, being careful to avoid assertions of primacy. 

Please avoid using ‘effect’, which should be used only if causality can be inferred, i.e., from an RCT. 

Author Summary:

In the final bullet point of ‘What Do These Findings Mean?’, please describe the main limitations of the study in non-technical language.

Introduction:

Line 104 - please temper assertions of primacy by adding ‘to the best of our knowledge’ or similar 

Methods:

Thank you for appending your analysis plan. Please cite this early in the Methods section. Changes in the analysis-- including those made in response to peer review comments-- must be identified as such in the Methods section of the paper, with rationale. If a reported analysis was performed based on an interesting but unanticipated pattern in the data, please be clear that the analysis was data-driven.

Please add the following statement, or similar, to the Methods: "This study is reported as per the REporting of studies Conducted using Observational Routinely-collected Data (RECORD) guideline (S1 Checklist)." When completing the checklist, please use section and paragraph numbers, rather than page numbers which will likely no longer correspond to the appropriate sections after copy-editing.

Line 205 - please refer to the specific supplementary file where further details can be found.

Results: 

Please provide a table showing the baseline characteristics of the study population (as Table 1).

Please report the number of individuals included in your study.

Please clarify whether CIs presented are 95% CIs.

Where ORs are presented, please specify the comparison group.

Discussion:

Please present and organize the Discussion as follows: a short, clear summary of the article's findings; what the study adds to existing research and where and why the results may differ from previous research; strengths and limitations of the study; implications and next steps for research, clinical practice, and/or public policy; one-paragraph conclusion.

Please remove all subheadings within your Discussion e.g. Limitations and other considerations

Figures:

Please define all abbreviations used in the figure legend of each figure.

Please consider avoiding the use of red and green in order to make your figure more accessible to those with colour blindness. 

Fig. 1, Fig S1, Fig 3, Fig S5 - Please clarify what is meant by the time in brackets on the label of your y axis.

Tables:

When a p value is given, please specify the statistical test used to determine it in the table legend.

Please define all abbreviations used in the table legend of each table.

References:

Please ensure that journal name abbreviations match those found in the National Center for Biotechnology Information (NCBI) databases, and are appropriately formatted and capitalised.

Please also see https://journals.plos.org/plosmedicine/s/submission-guidelines#loc-references for further details on reference formatting. For example, 6 authors should be named prior to ‘et al’.

Supplementary files: 

Please see https://journals.plos.org/plosmedicine/s/supporting-information for our supporting information guidelines. 

Comments from the reviewers:

Reviewer #1: Review of Manuscript Number: PMEDICINE-D-21-03163R1, "Uptake of infant and pre-school immunisations in Scotland and England during the COVID-19 pandemic: an observational study of routinely collected data"

Overall Impression

This study serves to describe the impact of COVID-19 lockdowns in England and Scotland on routine preschool immunizations. Using routinely collected data from Public Health Scotland and ImmForm in England, the authors found consistent increases in timely vaccination rates during the COVID-19 pandemic lockdown in Scotland for all five vaccine doses examined. In England, authors found decreases in the fraction of children who were up-to-date for their childhood vaccines during the lockdown period compared to 2019. 

This analysis contributes to a growing body of research quantifying the impact of the COVID-19 pandemic on routine childhood immunizations. While much of the research focuses on early phases of the pandemic and finds substantial disruption, this article provides an example (in Scotland) of a program experiencing increases in immunization during this period, the opposite effect.

Strengths of this manuscript include substantial data availability and disaggregation by vaccine dose, age, geography, and deprivation variables in Scotland. The data in England was more limited, however, and could not be compared directly to that of Scotland. Additionally this study was observational in nature and did not address the underlying drivers or reasons for these patterns and trends. 

I recommend that the editors publish this manuscript after minor revisions. Specific recommendations are included below. 

Major Issue

1. The authors describe that they use "aggregate binary logistic regression" with "vaccination status (vaccinated vs unvaccinated) as the dependent variable" (lines 192-94). It is unclear to me the definition of this method and the exact formulation used. If aggregate percentages were used, how were these values weighted in the analysis? Please provide additional detail so that this analysis could be replicated. 

Minor Issues

2. In Figure 1 the authors provide the annual uptake for 2019, monthly uptake for Jan and Feb of 2020, and weekly update for March to September 2020. While this plot is helpful, it masks any seasonal variation that may be expected in vaccine delivery. It would be helpful to see the monthly or weekly pattern of vaccine delivery in 2019 to compare to what was observed in 2020. The authors also mention this in their limitations (lines 354-59); this analysis fails to account for seasonality in vaccine delivery.

3. The y-axis on figure 4 makes it very hard to notice the small differences in the bars. Could you use a smaller y-scale (for example 75 to 100) or a different type of chart to visualize this data?

4. This manuscript provides consistent and convincing data to suggest that vaccine delivery significantly increased during the COVID-19 lockdown in Scotland. The authors also acknowledge that the nature of this data do not allow for detailed understanding of the precise reasons for that trend. For these results to be most useful, it would be important for clinicians and policy-makers to understand the precise drivers of success to apply them in settings where COVID-19 continues to disrupt routine health services or in order to build more robust and resilient vaccine delivery systems in other countries. The authors provide two primary hypotheses—A) that flexible work increased appointment attendance (lines 379-83) and B) that accessibility and pre-appointment reminders increased uptake (lines 384-93). It would be helpful to supplement this analysis with qualitative or contextual data to support these hypotheses or provide additional strategies that led to the success observed. I would also like to know if the strategies described in B) were unique to the COVID-19 lockdown period? What was the timeline for these interventions? The authors also provide some hypotheses for the decrease in England (lines 394-405) but do not explain why these factors were abated in Scotland. What were the specific program differences between England and Scotland that may have led to these differences in uptake?

5. The authors acknowledge in the results that the positive impact on vaccine uptake in Scotland was primarily in the "timeliness" of vaccines and that "the impact on longer term or final achieved uptake was more variable, possibly reflecting a ceiling effect on maximal uptake, for the earliest immunisations" (294-307). Based on this description and the results of this analysis, it appears that the COVID-19 lockdown may have promoted more prompt vaccinations but not overall vaccine coverage. Could the authors provide a quantitative analysis on the impact of prompt vaccination? Especially during the COVID-19 pandemic when pandemic-related behavior change such as reduced physical contact likely decreased the transmission of many infectious diseases, is vaccination a few weeks or even months earlier likely to have a big impact on the transmission of vaccine-preventable diseases?

Reviewer #2: 1. The results could be potentially biased when the underlying trend and seasonal patterns (e.g. if there is an increasing trend in the uptake, and or the update proportion decrease in the holiday season in December) are not controlled. Since weekly and/or monthly data are available, I would recommend to extend the study period (e.g. Add data of 2017 and 2018 ) and employ a more robust interrupted time series design in order to control for the underlying secular trend and seasonal variation in the pre-Covid-19 period. Although the author points out in the limitation that quarterly trends published for previous years including 2019 do not show major difference in uptake throughout the year and in fact show a gradual decline in uptake since 2015, I would suggest to analyze and graphically demonstrate the trend of the weekly/monthly uptake proportion for the years before the pandemic.

2. The author talked little about how they handle missing data. As shown in Figure 2, missingness (or does not missingness here represents not significant change? The author needs to clarify this) mainly exited in regions with higher baseline uptake proportion. This missingness not at random may confound the overall estimates.

3. I would suggest to the title of Fig 2 because it not only shows the baseline mean update proportion. What's the difference between "Absolute % change" and "% point change" ?

4. For figure 4, I would suggest to reset the limits of y axis or present the % of unvaccinated as the difference between the four periods in the current figure is hardly noticeable. 

5. For figure S1, I would suggest to add the geographic areas with missing data. 

6. The code is not available from the Github page. This needs to be updated accordingly.

7. The author may consider sensitivity analysis examining uptake within 2 and 6 weeks of eligibility

Reviewer #3: The authors assess the impact of the COVID-19 associated lockdowns in the UK and Scotland on pre-school vaccinations rates. Their results are reassuring in that they found that the coverage rates were higher in Scotland during the lockdown and while the UK did experience a decline, it was relatively small. My comments are provided in the attached draft manuscript and mostly focus on the the figures and tables. Several of the figures need considerable editing to be understandable and able to stand alone. I also suggest the authors considered omitting a few of the supplemental figures that aren't needed. Overall the manuscript is well written and contains relevant data. I hope the authors take the time to ensure their figures effectively highlight and clarify their results.

[LINK]

---

## [Decision Letter · Decision Letter 2]

10 Jan 2022

Dear Dr. McQuaid,

Thank you very much for re-submitting your manuscript "Uptake of infant and pre-school immunisations in Scotland and England during the COVID-19 pandemic: an observational study of routinely collected data" (PMEDICINE-D-21-03163R2) for review by PLOS Medicine.

I have discussed the paper with my colleagues and the academic editor and it was also seen again by three reviewers. I am pleased to say that provided the remaining editorial and production issues are dealt with we are planning to accept the paper for publication in the journal.

[LINK]

We look forward to receiving the revised manuscript by Jan 17 2022 11:59PM.   

Sincerely,

Louise Gaynor-Brook, MBBS PhD

PLOS Medicine

plosmedicine.org

Requests from Editors:

Abstract Methods and Findings:

Line 49 - please revise to ‘We conducted an observational study’ or similar

Line 65 - please define OR and CI at first use 

Author Summary:

In the final bullet point of ‘What Do These Findings Mean?’, please describe the main limitations of the study in non-technical language.

Results: 

Line 269 - please revise to “14.3% for”

Discussion:

Line 398 - please remove ‘in fact’

Line 401 - please remove ‘Although’ at the beginning of the sentence 

Line 433 - please revise to ‘suggesting’

Line 464 - please revise to ‘significantly’

Line 473 - please revise ‘vital given in’

Figures:

All figures - please specify country in the legend

Fig S1 - please indicate sections A and B within the figure itself

Fig S4 - dashed blue lines appear to be missing from the figure itself

Fig S2 - Please note that figures cannot be reproduced from sources that are not CC-BY. Please be sure to check the usage rights. 

Tables:

S2 Table - please specify country in the legend

Comments from Reviewers:

Reviewer #1: I am satisfied with the edits made to this manuscript in response to review and believe it should be accepted for publication.

Reviewer #2: We thank the author for answering reviewers's question and revising the paper.

1. The title and legends of all Figures are missing

Reviewer #3: The authors addressed all of my concerns in their revised manuscript.

[LINK]

---

## [Editor Report · Decision Letter 3]

17 Jan 2022

Dear Dr McQuaid, 

On behalf of my colleagues and the Academic Editor, Prof. Lars Åke Persson, I am pleased to inform you that we have agreed to publish your manuscript "Uptake of infant and pre-school immunisations in Scotland and England during the COVID-19 pandemic: an observational study of routinely collected data" (PMEDICINE-D-21-03163R3) in PLOS Medicine.

PRESS

Sincerely, 

Louise Gaynor-Brook, MBBS PhD 

Associate Editor, PLOS Medicine